# Structural Insights into TOR Signaling

**DOI:** 10.3390/genes11080885

**Published:** 2020-08-04

**Authors:** Lucas Tafur, Jennifer Kefauver, Robbie Loewith

**Affiliations:** 1Department of Molecular Biology, University of Geneva, 30 quai Ernest-Ansermet, CH1211 Geneva, Switzerland; Lucas.TafurPetrozzi@unige.ch (L.T.); Jennifer.Kefauver@unige.ch (J.K.); 2Swiss National Centre for Competence in Research (NCCR) in Chemical Biology, University of Geneva, Sciences II, Room 3-308, 30 Quai Ernest-Ansermet, CH1211 Geneva, Switzerland

**Keywords:** target of rapamycin, structural biology, cell growth homeostasis

## Abstract

The Target of Rapamycin (TOR) is a highly conserved serine/threonine protein kinase that performs essential roles in the control of cellular growth and metabolism. TOR acts in two distinct multiprotein complexes, TORC1 and TORC2 (mTORC1 and mTORC2 in humans), which maintain different aspects of cellular homeostasis and orchestrate the cellular responses to diverse environmental challenges. Interest in understanding TOR signaling is further motivated by observations that link aberrant TOR signaling to a variety of diseases, ranging from epilepsy to cancer. In the last few years, driven in large part by recent advances in cryo-electron microscopy, there has been an explosion of available structures of (m)TORC1 and its regulators, as well as several (m)TORC2 structures, derived from both yeast and mammals. In this review, we highlight and summarize the main findings from these reports and discuss both the fascinating and unexpected molecular biology revealed and how this knowledge will potentially contribute to new therapeutic strategies to manipulate signaling through these clinically relevant pathways.

## 1. Introduction

The *TOR* (*Target Of Rapamycin*) genes were first identified in a screen of yeast mutants that were resistant to the macrolide rapamycin [1], a compound produced by the soil bacterium *Streptomyces hygroscopicus* collected by bioprospectors from Easter Island in the 1960s [2]. The encoded Tor protein was to become the founding member of the family of Phosphatidylinositol 3-kinase-related kinases (PIKKs) [3] – Ser/Thr-protein kinases with curious sequence similarity to phosphatidylinositol-3 kinases (PIKs). Over the years, studies have revealed that Tor serves as a central hub in highly conserved signaling networks that govern various aspects of cell and organismal physiology. Accordingly, dysregulation of mTOR (mammalian or mechanistic TOR) signaling in humans has been implicated in a myriad of diseases including metabolic syndrome, cancer and neurodegenerative diseases [4]. 

Tor assembles into two distinct complexes (parentheses indicate mammalian nomenclature): rapamycin-sensitive (m)TORC1 and rapamycin-insensitive (m)TORC2, which have different subunit compositions and downstream targets (for a detailed review, see [4,5]). In recent years, in part due to improvements in cryo-electron microscopy (cryo-EM), there has been an explosion of available structural information for both (m)TOR complexes, related PIKKs, and (m)TORC1 regulators which we review here (Table 1 and Table 2). Overall, these structural studies are paving the way for a more detailed and mechanistic understanding of the function and regulation of both mTORC1 and mTORC2 and generating exciting drug discovery insights.

## 2. mTORC1 Signaling Network

Relative to mTORC2, rapamycin-sensitive mTORC1 signaling has thus far been easier to study. Many of its downstream targets and upstream regulators have been identified and characterized (Figure 1). When active, mTORC1 initiates a phosphorylation cascade involving a variety of substrates that collectively promote cell growth and inhibit autophagy [4]. This pro-growth regime involves the upregulation of protein synthesis [7], nucleotide synthesis [8], and lipid synthesis [9] as well as the promotion of aerobic glycolysis [10]. Meanwhile, mTORC1 also suppresses catabolism by limiting lysosome biogenesis and inhibiting autophagy [11]. Because mTORC1 activates an energy-intensive regime, it is tightly controlled. Several layers of regulation integrate signals of nutrient availability and cell stress to enhance or reduce, respectively, mTORC1 activity. The endogenous mTORC1 inhibitors proline-rich AKT substrate 40 kDa (PRAS40) [12] and DEP-domain-containing mTOR-interacting protein (DEPTOR) [13] bind directly to mTORC1 to reduce its activity. Another layer of mTORC1 regulation is its localization. mTORC1 must be recruited to the lysosome by the heterodimeric Ras-related GTPases Rag (RagA/B and RagC/D) which are themselves anchored to the lysosome by the Ragulator (LAMTOR1-5) complex [14]. Once at the lysosomal surface, mTORC1 can be allosterically activated through binding of another G-protein, Ras homolog enriched in brain (Rheb) [15,16].

Structures of (m)TORC1 and several complexes that regulate its activity have revealed new details about how this complex network of proteins is able to sense signals in the cellular environment and execute its pro-growth function. These structural insights are discussed now.

## 3. mTORC1 Structure

mTORC1 is composed of three defining subunits: mTOR, mLst8 (mammalian lethal with Sec13 protein 8) and Raptor (regulatory protein associated with mTOR) (Table 1). These subunits are arranged in a hollow rhomboid dimer containing two copies of each subunit and exhibiting C2 symmetry [17,18,19] (Figure 2A). At its N-terminus, mTOR has two contiguous HEAT (Huntington, EF3A, ATM, TOR) domains, the N-terminal HEAT (N-HEAT) domain and the middle HEAT (M-HEAT) domain. These are followed by the FAT (FRAP, ATM, TRRAP) domain, and finally the kinase domain (KD) at the C-terminus, which, like canonical eukaryote protein kinases, is formed by two lobes (N-ter and C-ter lobes) that form the active site cleft (Figure 2B). This domain architecture (an N-terminal α-solenoid region, followed by the FAT and KD) is conserved with other members of the PIKK family [3] (see Section 9). Three important elements are observed in the KD of (m)TOR (Figure 2B). The FKBP12-rapamycin Binding (FRB) domain is found in the N-ter lobe. The proximity of the FRB domain to the active site suggests that FKBP12-rapamycin binding inhibits mTORC1 primarily by sterically occluding the active site [20] (Figure 2B,C). The Lst8 binding element (LBE) is localized in the C-ter lobe, and as its name suggests, serves as a binding platform for Lst8. Finally, the FAT C-terminal (FATC) domain is situated at the very C-terminus of (m)TOR. Because the FAT and FATC domains are always observed together, it was thought that they interacted directly with each other [21]. However, in mTOR, the FATC forms an integral part of the KD but does not interact with the FAT domain [20].

The core of mTORC1 is composed of two mTOR molecules, each copy interacting with the other in a “ying-yang” manner. The N-HEAT folds into a “horn” or “spiral” structure, which interacts with the contiguous M-HEAT domain from the other monomer in the complex [18,19,22]. This interaction interface is also shared with the N-terminal CASPase-like domain of Raptor, which contacts both the N-HEAT domain of one mTOR molecule, and the M-HEAT domain and kinase domain of the other mTOR copy (Figure 2A). This architecture produces the characteristic rhomboid shape of the complex, with each copy of mLst8 peripherally bound to the LBE domain in the mTOR KD (Figure 2A,C). The N-terminal CASPase-like domain of Raptor is followed by a block of α-solenoid/armadillo repeats (ARM) in the middle, and a WD40 domain at the C-terminal end (Figure 2D). The C-terminal half protrudes out of the complex, enabling it to form other protein-protein interactions (see Section 4).

A TOR signaling (TOS) motif has been described in the two best-studied mTORC1 substrates, 4E-BP1 and p70S6K [23]. Binding of this motif was biochemically mapped to the N-terminal domain of Raptor [24]. Later structures showed that the TOS motif is located in a groove between the CASPase and α-solenoid domains of Raptor, approximately 65 Å away from the active site (Figure 2A,D) [17]. Whereas binding of 4E-BP1 to Raptor is essential for mTORC1-dependent phosphorylation [24,25,26], Raptor is not required, but rather plays a stimulatory role for p70S6K phosphorylation, presumably through increasing its local concentration [24,26]. Indeed, it has been shown that an mTOR variant lacking the N-terminus (comprising only the FAT and KD, residues 1376–2549) in complex with Lst8 is able to phosphorylate its substrates in the absence of Raptor [20], arguing for a TOS-independent mechanism of substrate recruitment. Consistently, a short fragment of p70S6K forms an amphipathic helix which interacts with the FRB domain (Figure 2C) and disruption of this helix or its interaction with the FRB domain affects mTOR-mediated phosphorylation of p70S6K in vitro [17]. Interestingly, other mTORC1 substrates such as Grb10, TFEB, Maf1, LIPIN and 4E-BP1 also contain amphipathic helices in proximity to their phosphorylation sites, and mutation of these strongly reduces their phosphorylation by mTOR in vitro [17]. Altogether, these results suggest a two-site mechanism of substrate recognition of mTORC1: one mediated by Raptor (TOS-dependent) and the other by binding to the FRB domain (TOS-independent). The latter could presumably involve a competition between FKBP12-rapamycin and substrates for the FRB site [17]. 

The mTORC1 inhibitor, PRAS40, has been shown to interact with both Raptor and the mTOR KD [12,27]. PRAS40 also harbors a TOS motif and can be phosphorylated by mTORC1, competing with 4E-BP1 and p70S6K [28]. Structural studies have shown that PRAS40 can bind both to Raptor via its TOS motif and to the mTOR FRB domain via an amphipathic helix, similar to p70S6K (Figure 2C) [17]. This helix is connected by a disordered segment to a beta-strand that interacts with mLst8 and stabilizes the interaction. Thus, PRAS40 inhibits mTORC1 activity by interacting with both of its substrate binding sites. In contrast, DEPTOR, which inhibits mTORC1 and mTORC2, interacts with the mTOR FAT domain [13], suggesting a different, yet to be defined, mode of inhibition. 

## 4. mTORC1 Activation: Rag and Rheb GTPases

The most widely shared view of mTORC1 activation involves its obligate recruitment to the lysosomal membrane through interaction with the Rag-Ragulator complex [14]. Ragulator is a pentameric complex composed of LAMTOR1-5 and is anchored to the lysosomal membrane via the N-terminal tail of LAMTOR1, which can undergo myristoylation and palmitoylation [29]. The two pairs of Rag GTPases, RagA and RagB, and, RagC and RagD function as heterodimers (RagA or RagB with RagC or RagD). The guanine nucleotide loading status of the Rags is determined downstream of signals initiated by dedicated nutrient sensors. When nutrient signals are permissive for growth, RagA/B are loaded with GTP and RagC/D with GDP, and this combination binds mTORC1, recruiting it to the lysosome [30].

Using a mutant variant that has a more stable interaction with mTORC1, two recent studies have reported the mode of interaction of RagA/RagC with mTORC1 (Figure 3A) [31,32]. RagA/RagC interacts with mTORC1 via the α-solenoid domain of Raptor through its N-terminal GTPase domains (G-domains), while their C-terminal roadblock domains (CRD) interact with each other and with the LAMTOR1 and LAMTOR2 subunits from Ragulator (Figure 3B). The selectivity for Raptor binding to the GTP-GDP conformation appears to be mediated by the mode of interaction between Raptor and RagA/RagC. On one hand, the main interaction surface involves the switch I and inter-switch regions of RagA, which are predicted to change conformation depending on whether they are bound to GTP or GDP [33]. On the other hand, insertion of a stretch of amino acids (denominated the “Raptor claw”, residues 916 to 937) into the RagA/RagC G-domain interface (Figure 3C), brings Glu935 of Raptor in close proximity to the GDP in RagC. This insertion is made possible because the space between the G-domains is widened compared to the structure of the free dimer [31]. Superimposition of the GTP-bound structure of RagC shows that in this conformation, there would be a clash between the RagC switch I and the tip of the claw, and the inter G-domain space would be narrower [32]. Modelling of different combinations of nucleotide loading in RagA and RagC revealed that the interactions observed are favored only in the active conformation. Thus, the interaction between Raptor and RagA/RagC is only possible when the G-domains are loaded with the GTP-GDP combination, which allow both the interactions between the Raptor α-solenoid and the switch I interface of RagA and the insertion of the Raptor claw into the widened inter-G-domain space. The structures do not show any conformational change in mTORC1 upon RagA/RagC binding, consistent with the notion that the Rag-Ragulator complex serves only indirectly in mTORC1 activation via recruitment to the lysosome.

Once recruited to the lysosomal membrane, subsequent activation of mTORC1 requires interaction with the other major GTPase regulator of mTORC1, Rheb. Unlike the Rags, Rheb can interact directly with the lysosomal membrane via its farnesylated C-terminal tail [34]. In metazoans, the guanine nucleotide of Rheb appears to be primarily, but not exclusively, dictated downstream of signals generated by growth factor receptors [4,35]. In this way, signals from both nutrients (via Rags) and growth factors (via Rheb) converge at the lysosome, and both are required for full mTORC1 activation [36]. The structure of the mTORC1-Rheb complex shows that Rheb binds both the HEAT and FAT domains of mTOR on an interface opposite to the active site (Figure 3D) [17].

Together, the structures described above suggest a model for the anchoring of the activated mTORC1 complex to the lysosomal membrane (Figure 3D) [31,32]. In this model, Rag-Ragulator anchors mTORC1 to the lysosomal membrane and restricts its movement, promoting its interaction with the transient pool of Rheb localized to the lysosome. The orientation of the anchored mTORC1 is such that the active site of each monomer faces away from the lysosomal membrane, being potentially accessible to substrates. Alternatively, the mTORC1-Rheb complex could dissociate and phosphorylate substrates which are in other cellular locations [34].

Binding of Rheb to mTOR requires a significant movement of the N-HEAT domain towards the M-HEAT domain (Figure 4A). This drags and shifts the FAT domain, promoting new interactions with the N-HEAT domain and closing the active site cleft. Ultimately, these changes bring the ATP phosphate groups closer to the catalytic and magnesium-coordinating residues in the active site (Figure 4B), producing a 30-fold increase in the catalytic rate of mTORC1 in vitro [17]. The conformational change induced by Rheb binding precludes the existence of a mixed mTORC1 dimer in which only one mTOR monomer is bound to Rheb (Figure 4C). It is interesting to note that if binding of Rheb occurs after mTORC1 has been recruited to the lysosome, the significant conformational changes in mTOR might destabilize the anchored super-complex. This would support the idea that the mTORC1-Rheb complex can leave the lysosome [34]. As mTOR in the mTORC1-RagA/RagC structure is in the inactive conformation, it remains to be seen if the mTORC1-Rheb-RagA/RagC-Ragulator structure exists in vivo or if it occurs only transiently.

The conformational changes observed upon mTORC1 activation also give insights into the effects of particular mutations. Hyperactivating and recurrent mutations in mTOR that have been associated with cancer appear to cluster to the FAT and KD (Figure 4D), and some have been shown to increase the catalytic activity of mTORC1 [17,37] and/or reduce the threshold for Rheb-induced activation in vitro [17]. Some mutations have also been shown to reduce binding to the negative regulator DEPTOR [37,38]. Interestingly, mutations in residue Y35 of Rheb increase p70S6K phosphorylation [38] and the Y35N mutant, in particular, has been found in five different tumor types [39]. The region of Rheb (residues ~20 to ~40) that interacts with the mTORC1 M-HEAT and FAT domains appears to be a mutational hotspot in different cancer types [39]. Because many cancer-associated recurrent mutations are clustered around the regions of mTOR that undergo some form of conformational change upon Rheb activation, it suggests that these changes are of pivotal importance for mTORC1 activity. Importantly, the mutations described above might have multiple, possibly synergistic, effects that contribute to the hyperactivation of mTOR [37].

## 5. Transducing Cellular Signals to mTORC1: Controlling the Rags and Rheb

The Rag GTPases have a very low intrinsic activity [40], which is exploited in vivo for tight control of their activity in response to upstream cues. In addition to providing an anchoring platform for binding to the lysosomal membrane, Ragulator has been proposed to act as a guanine exchange factor (GEF) for RagA/B [41]. However, it is not immediately apparent from the Rag-Ragulator structures how the proposed GEF activity of Ragulator would be manifest [32,42,43,44,45]; alternative RagA/B GEFs have been reported [46,47], and other data suggest that nucleotide exchange occurs in a spontaneous, uncatalyzed way [45].

Compared to nucleotide exchange, the action of the dedicated RagA/B GAP, named GAP activity toward Rags (GATOR1), is better defined but also still only partially characterized [48] (Figure 5A). GATOR1 is composed of three stable subunits: DEPDC5, NPRL2 and NPRL3. DEPDC5 constitutes the biggest of the three subunits, while NPRL2 and NPRL3 share a similar architecture, each having Longin domains in their N-terminal regions which interact with each other [48]. The cryo-EM structure of the GATOR1-RagA/RagC complex shows that DEPDC5 acts as a central protein that binds both NPRL2-NPRL3 through its structural axis for binding arrangement (SABA) domain and RagA/RagC through its steric hindrance for enhancement of nucleotidase activity (SHEN) domain (Figure 5A). Canonical GAPs insert an “arginine finger” or “asparagine/glutamine thumb” into the nucleotide binding pocket of their respective GTPase to stimulate GTP hydrolysis [49]. Interestingly, no such residue is observed in the GATOR1-RagA/RagC structure, and the previously proposed catalytic arginine, deduced from studies on the orthologous yeast protein Iml1 in the SEACIT complex, the yeast counterpart of GATOR1 (Table 2) [50], is far from the RagA GTP binding site. Point mutations in the DEPDC5 interface that contacts RagA (Figure 5A) increase the in vitro GAP activity towards RagA, which only requires NPRL2-NPRL3. In these experiments, DEPDC5 appears to increase the stability of the interaction between NPRL2-NPRL3, and, accordingly, in vivo, GATOR1 function requires all subunits [48]. Consistently, the catalytic arginine was later mapped to Arg78 in the NPRL2 Longin domain [51]. However, in the solved structure, this arginine is also located very far away and opposite to the RagA/RagC binding interface. Based on these results, the authors suggested that GATOR1 has two binding modes to RagA/RagC which are physically separated: one in which the GAP activity is inhibited by interaction with DEPDC5 (solved in the study), and one active conformation which has yet to be determined. 

In a similar vein, the RagC/D nucleotide-loading status is also controlled by a dedicated GAP, the folliculin (FLCN)-folliculin interacting protein (FNIP1/2) complex [52]. The FLCN-FNIP2 complex interacts directly with both RagA and RagC G domains [44,45] (Figure 5B). Akin to NPRL2-NPRL3, FLCN and FNIP2 also contain Longin domains at their N-termini that interact with each other. The Longin heterodimer inserts into the inter-G domain space and each subunit interacts in close proximity to the bound nucleotides—FLCN with RagA, and FNIP2 with RagC. The catalytic arginine finger is located in FLCN (Arg164) in a loop analogous to the loop containing Arg78 in NPRL2 (Figure 5C). Furthermore, as observed in the GATOR1-RagA/RagC complex, this conserved arginine is far away (~20 Å) from the RagC GTP. However, in the case of FLCN, Arg164 points towards the G domains. In this case, reaching the RagC-bound nucleotide would require a smaller local conformational change than in the case of GATOR1, which seemingly requires differential binding between DEPDC5 and NPRL2.

Comparison of the conformation of the RagA/RagC G domains also reveals different degrees of opening of the inter G-domain space, with the FLCN-FNIP2 bound structure presenting a much wider displacement of the G-domains compared to when either GATOR1 or Raptor is bound (Figure 5D). Intersubunit communication between G domains has been proposed to be important for locking the RagA/RagC heterodimer into a particular conformation [40]. Therefore, it appears that differential nucleotide binding also promotes changes in this space, which might affect interaction with specific binding partners, e.g., favoring one over another.

Analogously to the way in which the activator RagA/RagC GTPases are controlled by a dedicated GAP, Rheb activation is also tightly controlled by the Tuberous Sclerosis Complex (TSC), composed of TSC1, TSC2 and TBC1D7 [53]. TSC2 acts as a Rheb GAP, and thereby, TSC activation turns mTORC1 off [15,16]. The TSC is a very important hub in mammalian mTORC1 signaling as it integrates multiple cellular signals, from energy status to growth factor signaling [54,55]. The GAP activity of TSC2 has been mapped to its C-terminal region and a mechanism similar to Rap1-GAP using an “asparagine thumb” has been proposed [56,57]. While the structure of the N-terminal region, which mediates the interaction with TSC1, has been available for a few years [58], only recently was the crystal structure of the *Chaetomium thermophilum* GAP domain solved [59]. Based on structural modelling and previous biochemical data, the authors propose an important role for a key tyrosine residue in the Rheb switch I and an asparagine thumb in TSC2. A study currently under review has revealed, by cryo-EM, that the TSC complex consists of two copies of TSC1 and TSC2 each, and only one copy of TBC1D7 [60]. The functional insights derived from this structure are consistent with the previous TSC2 GAP structure [59].

## 6. Regulating the Regulators

Both GATOR1 [61] and FLCN-FNIP2 [62] appear to be regulated by phosphorylation by other Ser/Thr kinases, although the structural effects of phosphorylation on either of these complexes are not yet clear. GATOR1 is additionally regulated through interactions with two other protein complexes, KICSTOR and GATOR2. The lysosome-associated complex KICSTOR interacts with GATOR1 to recruit it to the lysosome in response to nutritional cues [63], while the GATOR2 complex appears to antagonize the GAP activity of GATOR1 in an unknown way [64].

Upstream of GATOR2 are dedicated amino-acid sensors. mTORC1 is responsive to, among other cues, the levels of specific amino acids in the cell, including leucine and arginine, in a Rag-dependent manner. Sestrin2 has been proposed to be the leucine sensor in mammalian cells [65] and CASTOR1, a sensor for arginine [66]. In the absence of leucine/arginine, Sestrin2/CASTOR1 is bound to GATOR2, and GATOR1 is free to inhibit mTORC1. When these amino acids bind to their respective complex, they dissociate from GATOR2, allowing it to inhibit GATOR1. This mechanism implies that leucine and arginine trigger a conformational change in Sestrin2/CASTOR1 that frees GATOR2. However, crystal structures of these sensor proteins have shown that there appears to be no significant conformational change between the apo- and bound forms [67,68] although for Sestrin2, there is still some controversy in whether any structure represents a true apo state [69]. Precisely how Sestrin2/CASTOR1 regulate GATOR2 in response to amino acid signals remain an open question. In addition, the structures and precise molecular mechanism of action of other molecular sensors, such as that of S-adenosylmethionine (SAMTOR), which interacts directly with GATOR1 [70], still remain to be elucidated. The lysosomal sodium-coupled amino acid transporter SLC38A9 has been proposed to signal arginine and cholesterol levels to mTORC1 [71,72]. Crystal structures of SLC38A9 provide a view of the cytosol-open state of the transporter and reveal its arginine binding site, but how SLC38A9 signals its loading state to the Ragulator-Rag complex remains to be determined [73].

## 7. Regulation of TORC1 in Budding Yeast

In the budding yeast *Saccharomyces cerevisiae*, TORC1 is composed of the Tor kinase (either Tor1 or Tor2, encoded by *TOR1* and *TOR2*, the two TOR paralogs), Kog1 (the counterpart of Raptor), Lst8 (mLst8) and the yeast-specific subunit Tco89. The TORC1 signaling network in fungi by and large share a common architecture with their mammalian counterparts, with the notable exception in *S. cerevisiae* which, unlike the fission yeast *Schizosaccharomyces pombe*, lacks *TSC1* and *TSC2* orthologs and does not appear to use its putative Rheb ortholog, Rhb1, to regulate TORC1 activity [74]. Nevertheless, TOR-related findings in yeast are generally relevant to our understanding of TOR signaling in bigger eukaryotes [75]. For example, studies have shown that both mTORC1 and budding yeast TORC1 have a very similar topology [22,76], highlighting the high degree of conservation between species. However, a high-resolution structure of ScTORC1 is still missing.

Whereas structural information on TORC1 is still limited, there are structural data on its regulators. In yeast, Gtr1 is structurally and functionally orthologous to RagA and B, and Gtr2 to RagC and D [77] (Figure 6A). The Gtr1/Gtr2 complex has a virtually identical architecture to RagA/RagC, consisting of a C-terminal roadblock domain (CRD) and an N-terminal GTPase domain (G domain). Similar to RagA/RagC, the Gtr1/Gtr2 complex interacts with a vacuole-anchored complex, the heterotrimeric EGO complex (composed of Ego1, Ego2 and Ego3) (Figure 6B), the functional and structural equivalent of Ragulator [78]. Ego1 and Ego3 cooperatively bind the CRDs of Gtr1/Gtr2 in a manner analogous to the binding of LAMTOR1 and LAMTOR2 to the CRDs of RagA/RagC. Gtr1/Gtr2 also bind the yeast equivalent of Raptor, Kog1 [46,79], although this specific binding architecture remains undefined. As expected due to the similarity with RagA/RagC, the G domains of Gtr1/Gtr2 also rotate and move relative to each other, changing the inter-G domain space according to guanine nucleotide binding [33] (Figure 6A). The similar orientations of Gtr1/Gtr2 when bound to the EGO complex and RagA/RagC when bound to Ragulator suggests a similar mode of interaction of these yeast GTPases with TORC1, although details of this interaction remain elusive. Yeast also share other regulatory components of the pathway, such as the SEACIT and SEACAT complexes (equivalents of GATOR1 and GATOR2), and Lst7-Lst4 (FLCN-FNIP2) [80,81]. However, structures of these have yet to be reported.

Interestingly, in *S. cerevisiae*, it has been shown that the activity of TORC1 in response to glucose availability is modulated by the reversible formation of a helical super-structure composed of TORC1 dimers (TORC1 organized in inhibited domains, TOROID) [76] (Figure 6C). A drop in glucose leads to TOROID formation in which the active site of Tor is occluded by an adjacent Kog1 α-solenoid domain, in a manner that shares striking resemblance to inhibition caused by binding of rapamycin-FKBP12 to the FRB domain (Figure 6C, right). When glucose levels increase, TOROIDs are disassembled via a Gtr1/Gtr2-dependent but molecularly-undefined mechanism, freeing TORC1 to phosphorylate its substrates. High-resolution structures of TORC1 alone or in complex with Gtr1/Gtr2 are anticipated to illuminate this mechanism. Regulation of the activity of TORC1 by assembly into super-structures, as has been observed in a growing number of enzymes [82,83], might offer an additional layer of regulation. It remains to be determined if such a super-structure exists in mammals, though the structural similarities between TORC1 in yeast and mammals suggest it is possible.

## 8. (m)TORC2 Structure

mTORC2 is composed of mTOR, mLst8, Rictor (Rapamycin-insensitive companion of Tor), mSin1 (mammalian stress-activated protein kinase SAPK-interacting protein) and Protor 1/2 (protein observed with Rictor). In yeast, only the Tor2 kinase is able to form TORC2 [84]; it consists of Avo3 (adheres voraciously to Tor2; counterpart of Rictor), Avo1 (counterpart of mSin1), Bit61/2 (counterpart of Protor 1/2) and Avo2 (yeast specific). The reason for only Tor2 being able to form TORC2 is not clear, particularly in the absence of high resolution structures of both TORC1 and TORC2. However, there is some evidence that suggests that a region in the N-terminal half of Tor2, called the major assembly specificity (MAS) domain, interacts with the other Tor2 in the dimer as well as with Avo2 and Avo3, and is therefore important for the stable interaction of Tor2 in TORC2 [85]. Inserting this MAS domain of Tor2 into Tor1, creates a chimera which is capable of associating with TORC2, but not TORC1. Future high resolution structures should help to better understand the structural differences between Tor1 and Tor2 that impact in their differential complex assembly.

Compared with the mTORC1 pathway, there is less functional and structural information available for mTORC2 signaling (Figure 1). mTORC2 was first shown to regulate cytoskeletal arrangements via the PKCα pathway [86,87]. mTORC2 also acts to suppress apoptosis, and to govern glucose homeostasis and anabolic metabolism [88,89,90]. The regulators of mTORC2 are also poorly defined. Regulation by the PI3K pathway is perhaps the best understood signal upstream of mTORC2, with phosphatidylinositol (3,4,5)-trisphosphate (PIP3) binding proposed to both recruit mTORC2 to the membrane and to stimulate its kinase activity [91]. In yeast, plasma membrane tension controls TORC2 activity in a way that may be recapitulated in mammalian cells [92,93,94,95].

During the last three years, models derived from cryo-EM images for budding yeast TORC2 [96] and for human mTORC2 [97,98,99] have been published. Overall, TORC2 and mTORC2 share a similar topology with each other as well as with mTORC1 (Figure 7A,B). The core of mTORC2 is formed primarily by interactions between mTOR monomers, as in mTORC1. However, comparison of the mTOR dimer in mTORC2 with that of mTORC1 reveals that the adjacent FAT domains are closer to each other, and thereby, the central hole of the complex is narrower (~11Å versus ~23Å) (Figure 7C). This appears to be due primarily to the presence of mSin1/Avo1 next to the FRB domain, which, together with Rictor/Avo3, “push” the KD and FAT domains towards the other mTOR molecule.

Rictor/Avo3 interacts with mTOR using a similar interface as Raptor, suggesting that their binding to mTOR is mutually exclusive. This provides an explanation for how mTOR can form two different complexes. However, the architectures of Rictor and Raptor are not identical; whereas the C-terminal region of Raptor protrudes out of the complex, the C-terminal domain of Rictor folds back on the complex and interacts with the mTOR FRB domain (Figure 7A) [99]. The cryo-EM reconstruction of TORC2 also suggested, based on unassigned but contiguous density, that the C-terminal region of Avo3 blocks the mTOR FRB domain (Figure 7D) [96]. Therefore, in both human and budding yeast, the C-terminal region of Rictor/Avo3 confers the rapamycin insensitivity of (m)TORC2 by interfering with the binding of FKBP12-rapamycin to the mTOR FRB domain. Consistently, in budding yeast, deletion of the C-terminal region of Avo3 renders TORC2 sensitive to rapamycin [100]. However, in humans, this region seems to be of greater/additional importance as, in contrast to yeast, C-terminal deletions of Rictor yield a non-functional complex [100]. It is worth mentioning that long-term rapamycin treatment can inhibit mTORC2 signaling in some cell types [101], most likely due to the binding to free mTOR molecules prior to their complex assembly.

Substrate recruitment to mTORC2/TORC2 appears to be mediated by mSin1/Avo1 via its conserved region in the middle (CRIM) domain [102,103]. In mTORC2, the N-terminal region of mSin1 adopts an extended conformation that spans over the active site cleft, interacting with both Rictor and mLst8. The distal N-terminus of mSin1 interacts with Rictor in a deep cleft between the two α-helical domains that organizes the N-terminal region of Rictor. mSin1 then extends over the active site cleft and wraps over mLst8 [99]. This might explain why mSin1 and mLst8 increase the stability of the interaction between Rictor and mTOR [97]. mSin1 also appears to stabilize interdomain interactions within Rictor and is therefore essential for mTORC2 assembly [97]. Crosslinking-mass spectrometry suggests a similar positioning of Avo1 in TORC2 [100]. In the yeast TORC2 reconstruction, weak density observed near the active site cleft was tentatively assigned to the Avo1 CRIM domain (Figure 7D) [96]. Weak density close to mLst8 was also assigned to the mSin1 CRIM domain in mTORC2, further suggesting a similar interaction in both complexes [99]. 

Finally, the recent structure of mTORC2 also revealed two small molecule binding sites: the “A-site” (that binds ATP) located in Rictor and the “I-site” (that binds inositol hexakisphosphate; InsP6) located in the mTOR FAT domain [99]. The A-site is a previously uncharacterized region located far away from the active site and is mTORC2-specific as it is located in Rictor. In contrast, the I-site was also observed in SMG1, another member of the PIKK family (see Section 9), and was also previously observed in mTOR [104]. While the function of the A-site is still elusive, binding of InsP6 appears to play a role in stabilizing the mTOR structure.

## 9. Beyond Cell Growth: Tor as a PIKK

As mentioned previously, mTOR is the founding member of the PIKK family. These proteins were initially misclassified as PIKs but now have emerged as an important protein family with a wide range of functions in the cell. There are five known PIKKs in addition to mTOR: DNA-PKcs, ATR and ATM (involved in DNA damage repair), SMG1 (involved in mRNA decay) and TRRAP (involved in chromatin remodeling) [3]. All PIKKs have Ser/Thr protein kinase activity except TRRAP, which is a pseudokinase, lacking any enzymatic activity [105,106]. Four PIKKs are conserved from budding yeast to mammals (TOR/mTOR, Tel1/ATM, Mec1/ATR and Tra1/TRRAP). Structures of all PIKKs have been reported and all share a similar domain architecture with mTOR: a variable N-terminal HEAT repeat region, followed by the FAT domain and the conserved KD, and a short FATC domain at the end - the FAT-KD-FATC domains are also called collectively “FATKIN” (Figure 8A). In all structures, the FAT domain links the divergent N-terminal α-solenoid region to the KD, thereby serving as a structural transducer of conformational changes occurring at the N-terminus. As the sequence identity of the N-terminal region is very low between PIKKs, this region appears to play a role in establishing protein-protein interactions with specific protein regulators. For instance, similar to Rheb binding to mTOR, binding of DNA/Ku70/80 to DNA-PKcs (which together form the DNA-PK complex) has been proposed to stimulate DNA-PKcs activity via an allosteric mechanism that involves the movement of the N-HEAT towards the FAT domain, leading to conformational changes in the KD [107]. Regulators and binding partners of other members of the family, such as those for ATR, ATM and SMG1, also bind to the N-HEAT [104,108,109] (Figure 8A). Therefore, the control of the activity of the PIKKs by their respective regulatory proteins might involve an allosteric mechanism mediated by binding to the N-HEAT domain, which can also serve as a platform for binding of specific proteins. 

The two-lobed PIKK KD presents several conserved elements: the ATP loop in the N-lobe, and the catalytic and activation loops, as well as the PIKK-regulatory domain (PRD), in the C-lobe [110] (Figure 8B). The PRD consists of two conserved helices (kinase (K) α9 and Kα10) connected by a divergent loop, termed the PRD insert (PRD-I) [110]. In mTOR, deletion of the PRD hyperactivates mTORC1, particularly towards 4E-BP1 and only slightly for p70S6K [111,112]. The mTOR PRD-I has only been observed in the cryo-EM structure of the thermotolerant yeast *Kluyveromyces marxianus* (KmTor) [22] and is disordered in all other reconstructions. In the yeast orthologue of ATR, Mec1, the PRD-I is an essential part of the dimerization interface and also blocks the activation loop [113], although this was not observed in the human structure [108]. Similarly, in the human DNA-PKcs structure, the PRD-I (denoted Hairpin 2) forms part of a network of hairpins that shield the activation loop and block substrate access [114] (Figure 8B). Therefore, it appears that the PRD regulates the activity and/or activation of PIKKs by modulating access to the active site. Interestingly, it has been recently proposed that the PRD could act as a pseudosubstrate, at least in some PIKKs such as SMG1 and ATM [115]. Accordingly, known phosphorylation sites of mTOR that are coupled to mitogen signaling are located in this region (Thr2446 and Ser2448) [112]. Together, the available structures suggest that the activity of the PIKKs can be regulated by changes in the conformation of specific active site elements such as the PRD. This would offer another layer of regulation on top of the binding of regulators to other parts of the protein. 

The PIKKs SMG1, Tra1 and DNA-PKcs (but not ATM and ATR) each have an alpha-helical bundle in the KD N-lobe, reminiscent of the FRB domain of mTOR. However, the relative positioning of this bundle in Tra1 is tilted towards the equivalent of the LBE domain in the C-lobe, which occludes the putative active site. This, coupled with the lack of conservation of the catalytic residues, likely explains the lack of kinase activity of Tra1/TRRAP [116]. However, a similarly closed active site is also observed for the crystal structure of DNA-PKcs, which strongly resembles the overall structure of Tra1. Interestingly, this conformation appears to represent an inactive state in which the active site is blocked by several elements from the KD, thereby requiring a large conformational change for activation [114]. 

The cellular assembly of multi-subunit protein complexes is a major feat that requires the concerted action of chaperones and proteins that aid in both proper subunit folding and inter-subunit interactions. This is particularly important for PIKKs, given their complex topology. Tel2 was identified as an essential protein needed for PIKK stabilization in vivo [117]. Tel2 binds directly to Tti1 and Tti2, and the Tel2-Tti1-Tti2 (TTT) complex serves as a co-chaperone for Hsp90 [118]. Tel2 appears to bind part of the HEAT repeats of mTOR and ATM early after their synthesis [117] and is essential for mTORC1/2 assembly [119]. A role for R2TP, a conserved multi-protein complex involved in diverse cellular processes [120], in stabilization of PIKKs (particularly mTOR and SMG1) has also been demonstrated [121,122] and might link energy status to mTORC1 function independently of the Rags [123]. 

Beyond mTOR, proper folding of the other mTORC1 subunits is also required. Recently, it has been shown that both mLst8 and Raptor bind to the chaperonin CCT via their β-propeller/WD40 domains, an interaction which is required for proper folding and mTORC1 assembly [124]. This interaction was not detected for mTOR, Rictor or mSin1. The cryo-EM structure of the CCT-mLst8 complex showed that the β-propeller of mLst8 was already folded in the CCT cavity, probably representing a late-intermediate step in the folding process and ready to be released for binding to mTOR. Therefore, assembly of mTORC1 requires the concerted action of CCT (for proper C-terminal folding of Raptor and mLst8) and the Hsp90 TTT-R2TP complex (for mTOR folding). 

## 10. Perspectives

The structural information presented in this review has significantly increased our knowledge of the molecular mechanisms by which mTOR activity is controlled. In particular, the structures of multiple PIKKs have revealed that findings for one member of the family can help in understanding of the regulation and function of other members of the family. However, many questions remain. For example, structures of many important TORC1 regulators including GATOR2, SAMTOR and KICSTOR remain to be reported, as well as regulators of TORC2 [92]. In a similar vein, the mechanisms through which yeast TOROID formation is regulated remains mysterious. Specifically, the role of the Gtrs is still poorly defined. It is tempting to speculate that in yeast, and possibly in mammals, the Gtrs (Rags) act directly to inhibit the formation of TOROIDs. Future structures of yeast TORC1 in complex with the Gtrs will help in elucidating this question. The identification of TOROIDs per se additionally invokes the obvious question: do other PIKKs form higher-order filaments? As in other fields, new goals from a structural biology perspective will include aiming to reconstitute bigger complexes in order to understand the interplay among regulators, as well as visualizing structures in situ by taking advantages of recent developments in cryo-electron tomography and correlative-light electron microscopy [125].

Dysregulation of mTORC1 [4], mTORC2 [126], and other PIKKs [127] is well known to be associated with various diseases. As our understanding of the molecular mechanisms governing the function of these big complexes develops, this knowledge could contribute to emerging therapies. In the case of mTORC1 signaling, for example, an interesting therapeutic target could be the inter G domain space of the Rag GTPases, which serves as an anchoring platform for mTORC1 as well as regulatory factors. Perhaps a small molecule or cell-penetrant peptide that mimics the raptor claw could effectively prevent lysosomal recruitment of mTORC1 and thus diminish mTORC1 signaling in a therapeutically useful manner. This could have value, for example, in the treatment of inherited focal epilepsy, which is caused by mutations in DEPDC5 and tends to be drug-resistant [128] or for by-passing the paradoxical increase in cell growth sometimes observed with rapamycin and its analogs [129]. In a similar vein, small molecules that antagonize the binding of Rheb to mTORC1 could find utility in the treatment of tuberous sclerosis complex or in cancers in which the Rheb axis has been corrupted. Inhibition of mTORC2 signaling may also be clinically useful, particularly in tumors driven by hyperactivation of PI3K. To this end, higher resolution / more complete structures of mTORC2 are required as well as a better understanding of its activation by membrane tension and other regulators. Lastly, lessons from other PIKKs along with a clearer understanding of the folding of these huge enzymes and their assembly into their obligate complexes may provide additional vantage points for therapeutic intervention. Given the novelty of the recently described PIKK and associated structures it is difficult to predict what the future will bring but, whatever it does, it promises to surprise and captivate.

## Figures and Tables

**Figure 1 genes-11-00885-f001:**
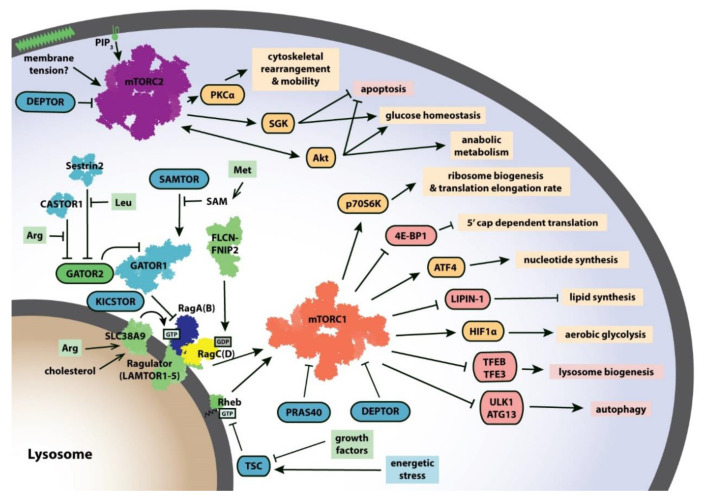
mTORC1 and mTORC2 integrate cellular signals to promote cell growth. Cellular signals including growth factors, amino acid availability, and energetic status are communicated through a complex regulatory network that impinges on mTORC1 and mTORC2 to produce a variety of pro-growth outputs. Targets of mTORC1 and mTORC2 that are positively (shown in goldenrod) or negatively (shown in pink) regulated are listed with their downstream effects. Several known upstream regulators of mTORC1 and mTORC2 are shown (positive regulators in green, negative regulators in turquoise). Structures of mTORC1 (shown in orange, PDB: 6BCX), mTORC2 (shown in purple, PDB: 5ZCS), RagA/RagC (shown in navy/yellow) with Ragulator (PDB: 6U62), Rheb (PDB: 6BCU), FLCN-FNIP2 (PDB: 6ULG), and GATOR1 (PDB: 6CES), Sestrin2 (PDB: 5DJ4) and CASTOR1 (PDB: 5GT8) have provided new insights into how these proteins function and interact with one another. Abbreviations: 4E-BP1, 4E-binding protein 1; Arg, arginine; CASTOR, cellular arginine sensor for mTORC1; DEPTOR, DEP-domain-containing mTOR-interacting protein; FLCN, folliculin; GATOR, GAP activity towards the Rags; HIF1α, hypoxia inducible factor 1α; Leu, leucine; Met, methionine; P70S6K, p70 S6 kinase 1; PIP3, phosphatidylinositol (3,4,5)-trisphosphate; PKCα, protein kinase Cα; PRAS40, proline-rich AKT substrate 40 kDa; SAM, S-adenosylmethionine; SAMTOR, S-adenosylmethionine sensor; TFE3, transcription factor E3; TFEB, transcription factor EB; TSC, tuberous sclerosis complex; ULK1, unc-51-like autophagy-activating kinase 1.

**Figure 2 genes-11-00885-f002:**
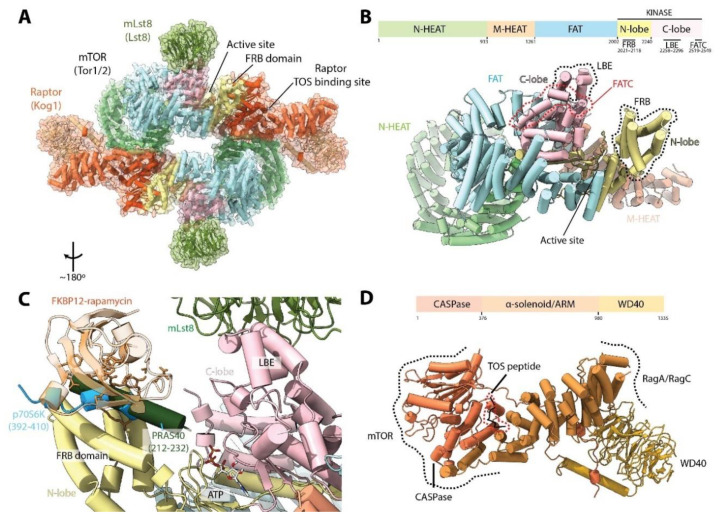
Structure of mTORC1. (**A**) Structure of mTORC1 (PDB:6BCX). Yeast nomenclature, in parenthesis, is shown below mammalian protein names. Two copies of each subunit are present in the complex. (**B**) Domain organization of mTOR. Scheme is not to scale. The LBE, FATC, and FRB elements present in the KD are indicated. (**C**) The mTOR active site. The active site cleft is formed by two lobes (N-ter and C-ter), and the FRB domain is located next to it. Binding of the FKBP12-rapamycin complex further restricts access to the active site. The binding site of a short segment from p70S6K (PDB. 5WBH), a helix from PRAS40 (based on PDB:5WBU) and the FKBP12-rapamycin complex (based on PDB:1FAP) to the N-lobe are shown. (**D**) Domain organization of *Arabidopsis thaliana* Raptor (PDB:5WBK). Scheme is not to scale. The TOR substrate (TOS) binding site is located in a groove between the CASPase and α-solenoid domains. Raptor interacts with mTOR via its N-terminal CASPase domain and with RagA/RagC via its α-solenoid domain (see Section 4). Regions in-between domains are omitted for clarity.

**Figure 3 genes-11-00885-f003:**
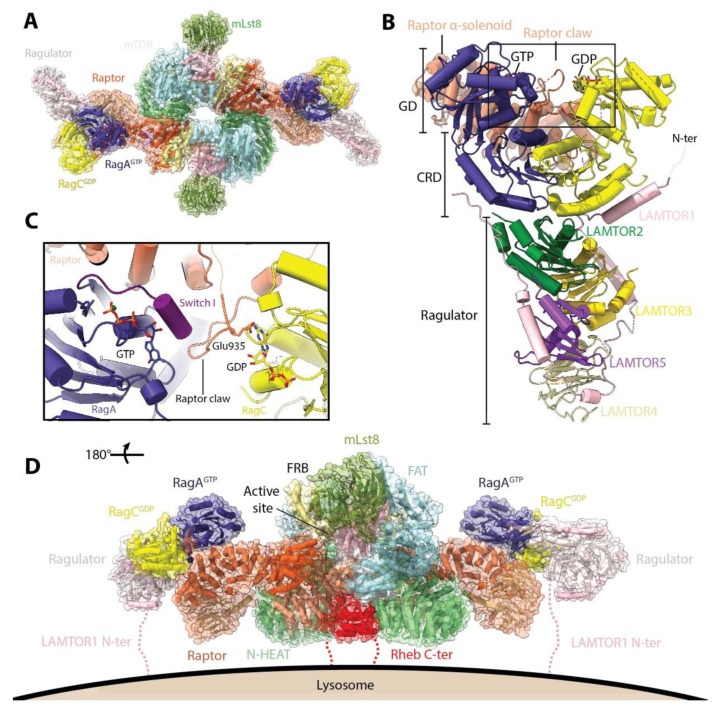
Structure of mTORC1 regulators Rag-Ragulator and Rheb. (**A**) Model of mTORC1 bound to RagA/RagC and Ragulator (based on PDB: 6BCX, 6SB0, and 6U62). The RagA/RagC heterodimer interacts with mTORC1 via Raptor. (**B**) Structure of Raptor-RagA/RagC-Ragulator super complex (PDB: 6U62). The domains of RagA/RagC are indicated: GD, G-domains; CRD, C-terminal roadblock domains. Only the raptor α-solenoid is shown for clarity. Ragulator binds to RagA/RagC via their CRD, while Raptor binds to the GD on the opposite end of the heterodimer. (**C**) Close-up of the interaction between Raptor claw and RagA/RagC. Raptor residue Glu935 comes into close proximity to the RagC-bound nucleotide. (**D**) Model of activated mTORC1 anchored to the lysosomal membrane. Model was constructed as in (**A**) but using the mTORC1-Rheb structure (PDB: 6BCU). The complex is anchored via both the N-terminal tail of LAMTOR1 and the C-terminus of Rheb, both of which undergo lipid modifications.

**Figure 4 genes-11-00885-f004:**
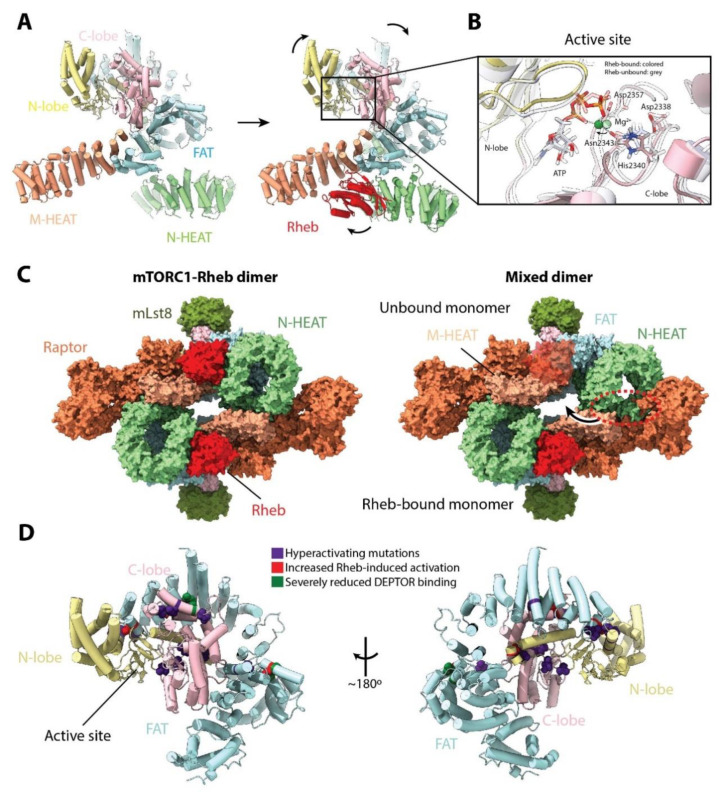
Rheb-induced mTORC1 activation. (**A**) Conformational changes in mTOR upon Rheb binding (PDB: 6BCX and 6BCU for unbound and bound mTOR, respectively). Binding of Rheb shifts the position of the mTOR N-HEAT domain towards the M-HEAT domain, which drags the FAT domain towards the N-HEAT domain. These changes align the catalytic residues in the active site. (**B**) Comparison between the active site of mTOR and mTOR-Rheb. (**C**) Comparison of the mTORC1-Rheb complex (**left**) with a mixed mTORC1 complex in which only one monomer is bound to Rheb (**right**). The conformational change induced by Rheb binding precludes the existence of a mixed dimer, as it would produce significant clashes (shown in a dotted red circle). (**D**) Cancer-associated hyperactivated mutations in mTOR mapped on the FAT and KD structure. View as in panel (**A**). Those mutations which have additionally been shown to reduce the Rheb-induced activation threshold are shown in red, whereas those that significantly reduce DEPTOR binding are shown in green. Note that not all mutations have been tested for these characteristics and they might have additional properties.

**Figure 5 genes-11-00885-f005:**
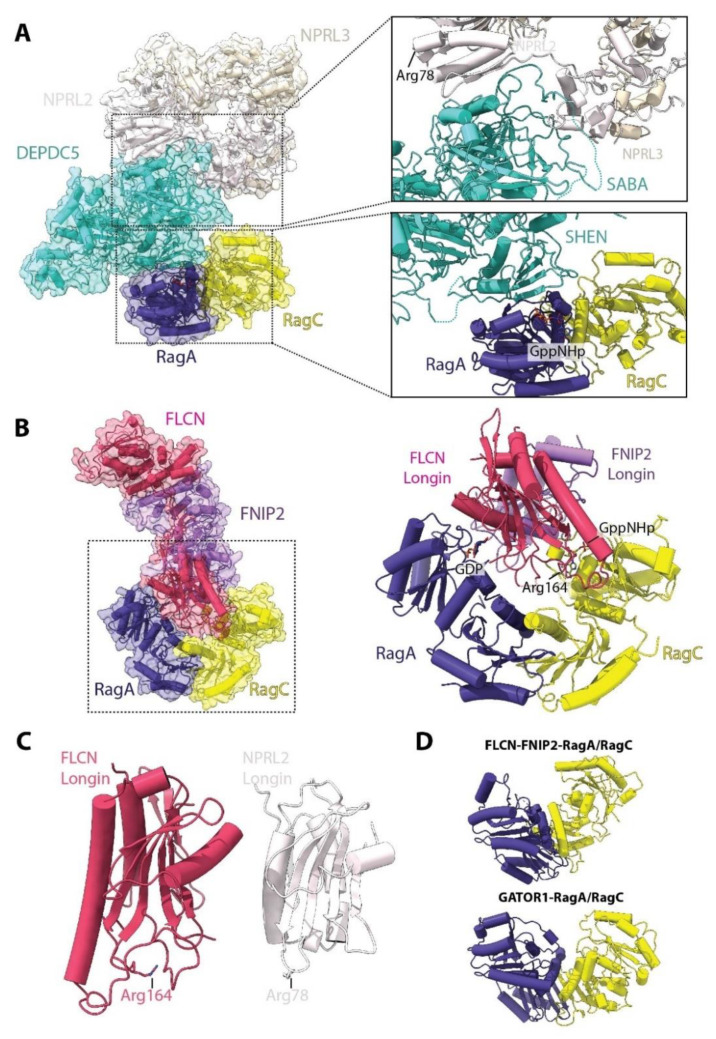
Structure of the RagA and RagC GAPs GATOR1 and FLCN-FNIP2. (**A**) Structure of the GATOR1-RagA/RagC (PDB:6CES) complex DEPDC5 interacts with RagA via its SHEN domain and with NPRL2 via its SABA domain. The catalytic Arg78 of NPRL2 is shown mapped to the structure, being far away from the RagA nucleotide binding site. (**B**) Structure of the FLCN-FNIP2-RagA/RagC complex (PDB:6ULG). FLCN-FNIP2 interacts with RagA/RagC via their Longin domains. The catalytic Arg164 of FLCN is indicated and is located distant from the RagC nucleotide binding site. (**C**) Comparison of the Longin domains of FLCN and NPRL2, the subunits with GAP activity from FLCN-FNIP2 and GATOR1, respectively. The position of the catalytic arginine for each is shown. (**D**) Comparison of the conformation of RagA/RagC when bound to FLCN-FNIP2 (**top**) or GATOR1 (**bottom**). The space between GD is the widest when bound to FLCN-FNIP and similarly smaller when bound to GATOR1 or Raptor.

**Figure 6 genes-11-00885-f006:**
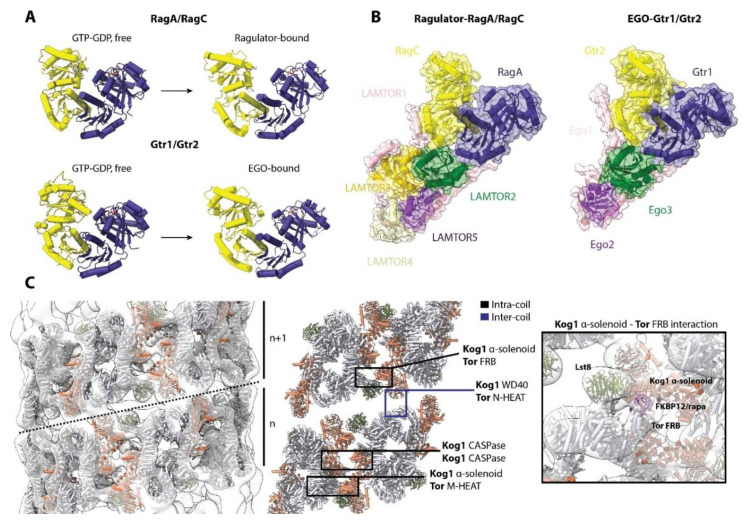
Structural conservation between yeast and humans. (**A**) Comparison of the structures of RagA/RagC (PDB: 6S6A) and Gtr1/2 (PDB:4ARZ) complexes, with or without Ragulator for RagA/RagC, and EGO for Gtr1/2. Both GTPases are able to rotate and move their G-domains and alter the inter-G domain expanse. (**B**) Comparison of the Ragulator-RagA/RagC complex (PDB:6U62) and the EGO-Gtr1/2 (PDB: 6JWP) complex. Both complexes show a very similar architecture: Ragulator interacts with the CRD of RagA/RagC via LAMTOR1 and LAMTOR2, while EGO interacts with the CRD of Gtr1/Gtr2 via Ego1 and Ego3. (**C**) Structure of the TORC1 TOROID (EMD-3814). The intra- (black) and inter-coil (blue) interactions are indicated, with a focus on the Kog1 α-solenoid—Tor FRB interaction that occludes the FRB domain and active site. The structure of mTORC1 (PDB:6BCX) fitted in the cryo-EM map is shown.

**Figure 7 genes-11-00885-f007:**
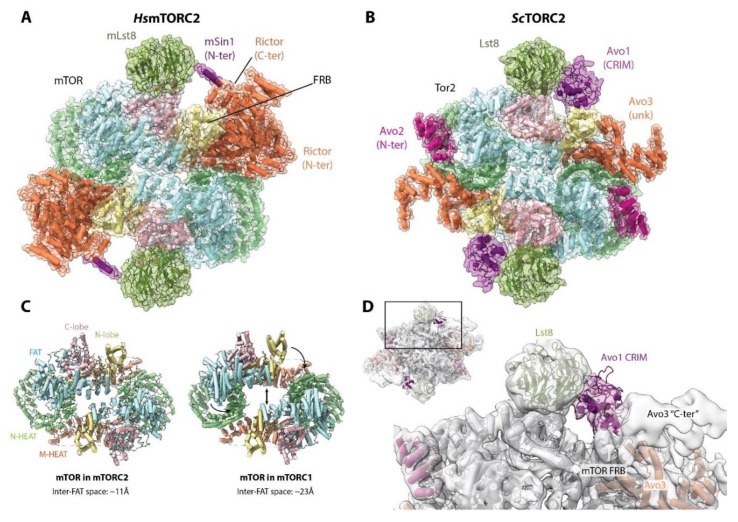
mTORC2 and TORC2 structures. (**A**) Structure of mTORC2 (PDB:5ZCS). (**B**) Structure of TORC2 (PDB:6EMK). The domains of each protein assigned to the low resolution densities are indicated, as well as the unassigned residues (unknown; unk). Two copies of each subunit are present in the complex. The position of the Rictor C-ter is based on the higher resolution structure of mTORC2 [99]. Both complexes have a remarkably similar architecture, and the overall conformation is also comparable to that of mTORC1. (**C**) Comparison of the conformation of mTOR in mTORC2 with that in mTORC1. The central hole of mTORC2 is narrower due to the shift in the position of the FAT domains. (**D**) Extra, unassigned density observed in the cryo-EM map of TORC2 agrees with the position of the Rictor C-terminal domain in mTORC2, suggesting that it corresponds to the C-terminal region of Avo3. A density, observed at low threshold in the yeast TORC2 map, next to the active site pocket and attributed to the Avo1 CRIM domain, is indicated.

**Figure 8 genes-11-00885-f008:**
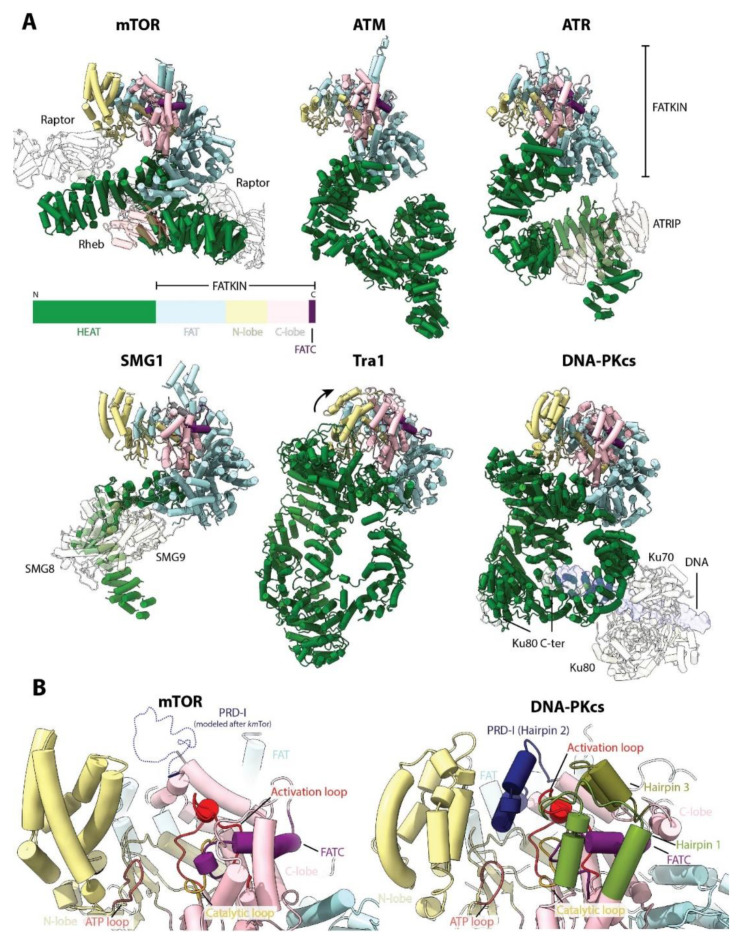
Compilation of PIKK structures. (**A**) Comparison of the structures of: *H. sapiens* mTOR (PDB: 6BCX), *H. sapiens* ATM (PDB:5NP0), *H. sapiens* ATR (PDB:5YZ0), *H. sapiens* SMG1 (PDB:6SYT), *S. cerevisiae* Tra1 (PDB:5OJS) and *H. sapiens* DNA-PKcs (PDB:5LUQ and 5Y3R). The conserved domain architecture is indicated schematically (not to scale). All PIKKs share a similar structure, where the N-terminal HEAT domain is the most variable. Other proteins that form part of the complex and/or activate the kinase activity are shown transparent. Tra1 forms part of the bigger SAGA and NuA4 complexes but does not have catalytic activity. For dimeric complexes (mTORC1, ATM), only one monomer is shown for clarity. (**B**) Comparison of the active site of mTOR (active) and DNA-PKcs (inactive). Conserved elements are indicated. The active site elements can undergo conformational changes in response to upstream cues to further control the activity of the PIKKs.

**Table 1 genes-11-00885-t001:** Target Of Rapamycin Complex 1 and 2 (TORC1 and TORC2) components in *Saccharomyces cerevisiae* and humans. Mass for (m)TORC1/2 assumes two copies of each subunit per complex. The names of the complete, functional complexes are indicated with a °. The mass of mTORC1 does not include PRAS40 and DEPTOR, and that of mTORC2 does not include DEPTOR. HUGO gene nomenclature is shown below in parenthesis for proteins for which the common name differs.

*Saccharomyces cerevisiae*	Molecular Weight (kDa)	*Homo sapiens*	Molecular Weight (kDa)
TORC1°	1164	mTORC1°	948
Tor1 or Tor2	281	mTOR	289
Kog1	178	Raptor *(RPTOR)*	149
Lst8	34	mLst8	36
Tco89	89	-	-
-	-	PRAS40 *(AKT1S1)*	27
-	-	DEPTOR	46
TORC2°	1436	mTORC2°	1152
Tor2	281	mTOR	289
Avo1	131	mSin1 *(MAPKAP1)*	59
Avo2	47	-	-
Avo3	164	Rictor	192
Lst8	34	mLst8	36
Bit61 or Bit2	61	Protor1 or Protor2 *(PRR5)*	42 or 41
-		DEPTOR	46

**Table 2 genes-11-00885-t002:** (m)TORC1 regulators in *Saccharomyces cerevisiae* and humans. The names of the complete, functional complexes are indicated with a °. The stoichiometries of SEACAT and GATOR2 components have not yet been determined. The Rheb homolog in *S. cerevisiae* is not implicated in TORC1 regulation [6] and thus omitted for clarity. The molecular weight of the multi-protein complexes is shown only for those for which there is a structure available. HUGO gene nomenclature is shown below in parenthesis for proteins for which the common name differs.

*S. cerevisiae*	Molecular Weight (kDa)	*H. sapiens*	Molecular Weight (kDa)
EGO complex°	121	Ragulator-Rag°	148
Gtr1	36	RagA or RagB *(RRAGA or RRAGB)*	37 or 43
Gtr2	39	RagC or RagD *(RRAGC or RRAGD)*	44 or 46
Ego1	20	LAMTOR1/p18	18
Ego2, possibly Ego4	8, 11	LAMTOR5/HBXIP	10
Ego3	18	LAMTOR2/p14	14
-	-	LAMTOR3/MP1	14
-	-	LAMTOR4/C7orf59	11
-	-	Rheb	20
-	-	TSC°	696
-	-	TSC1/Hamartin	130
-	-	TSC2/Tuberin	201
-	-	TBC1D7	34
SEACIT°		GATOR1°	289
Iml1	182	DEPDC5	181
Npr2	70	NPRL2	44
Npr3	130	NPRL3	64
SEACAT°		GATOR2°	
Sea4	118	Mios	99
Sea2	149	Wdr24	88
Seh1	39	Seh1L	40
Sec13	33	Sec13	36
Sea3	131	Wdr59	110
Lst7-Lst4°		FLCN-FNIP2°	186
Lst7	28	FLCN	64
Lst4	93	FNIP2	122

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
