# Peer review of "Structural Insights into TOR Signaling"

_genes, 2020, doi:10.3390/genes11080885_

Round 1
Reviewer 1 Report
Signalling via TORC1 – and to a lesser extent via TORC2 – is a highly dynamic and topical field of research. Tafur and colleagues review current insights into structural aspects of TOR signalling. They provide a comprehensive and up-to-date review, which will be appreciated by many researchers in the signalling field. The review covers most relevant aspects, highlights and acknowledges the relevant literature, and is well written. Besides focussing on the regulation (mostly of TORC1), it also emphasizes interesting similarities with other PIKK family members.
The review is a very nice piece of work and merits publication. Congratulations!
There are some minor issues that should be addressed:
- Figure 1: Some “activation arrows” and “inhibition symbols” are confusing. ULK1 activates autophagy; therefore, there should be an “activation arrow” between ULK1 and autophagy. The same for TFEB and lysosome biogenesis. By contrast, Lipin-1 inhibits lipid biosynthesis. Furthermore, the inhibition of GATOR1 by GATOR2 should also be indicated.
- line 93: see section 9 (not the last section).
- Table 1: Add “Hamartin” to TSC1.
- Figure 3D: The connection of LAMTOR1 N-ter should be to Ragulator, not to Raptor (on the right).
- line 308: Figure 5D
- line 341: to recruit it to the lysosome
- line 388: Lst7-Lst4 (instead of Lst4-Lst7). Alternatively, FNIP2-FLCN (instead of FLCN-FNIP2).
- line 418: Why is only Tor2 able to form TORC2? Reference 78 is an "invalid citation".
Besides these few amendments, I would like to suggest a change in wording. Rapamycin is often called “antibiotic” (line 27). This is certainly correct in the broad sense of the term “antibiotic”. However, in a more narrow sense, “antibiotic” is commonly used as a synonym for “antibacterial”. Therefore, the use of “antibiotic” in the context of rapamycin may be misleading.
On a final note, what could be added to this very thorough review are some thoughts on the evolutionary conservation (but this is probably beyond the scope).
Author Response
Reviewer 1
Replies are colored.
Signalling via TORC1 – and to a lesser extent via TORC2 – is a highly dynamic and topical field of research. Tafur and colleagues review current insights into structural aspects of TOR signalling. They provide a comprehensive and up-to-date review, which will be appreciated by many researchers in the signalling field. The review covers most relevant aspects, highlights and acknowledges the relevant literature, and is well written. Besides focussing on the regulation (mostly of TORC1), it also emphasizes interesting similarities with other PIKK family members.
The review is a very nice piece of work and merits publication. Congratulations!
We thank the reviewer for his positive comments.
There are some minor issues that should be addressed:
- Figure 1: Some “activation arrows” and “inhibition symbols” are confusing. ULK1 activates autophagy; therefore, there should be an “activation arrow” between ULK1 and autophagy. The same for TFEB and lysosome biogenesis. By contrast, Lipin-1 inhibits lipid biosynthesis. Furthermore, the inhibition of GATOR1 by GATOR2 should also be indicated.
We thank the reviewer for pointing this out and apologize for the oversight. The figure has now been corrected.
- line 93: see section 9 (not the last section).
The line has been edited accordingly.
- Table 1: Add “Hamartin” to TSC1.
Hamartin has been added.
- Figure 3D: The connection of LAMTOR1 N-ter should be to Ragulator, not to Raptor (on the right).
We have edited the figure accordingly.
- line 308: Figure 5D
The line has been edited accordingly.
- line 341: to recruit it to the lysosome
The line has been edited accordingly.
- line 388: Lst7-Lst4 (instead of Lst4-Lst7). Alternatively, FNIP2-FLCN (instead of FLCN-FNIP2).
The line has been edited accordingly. Additionally, we have updated the Table including these proteins.
- line 418: Why is only Tor2 able to form TORC2? Reference 78 is an "invalid citation".
We apologize for missing this. It has now been edited accordingly and added to the text as follows (Line 436):
“The reason for only Tor2 being able to form TORC2 is not clear, particularly in the absence of high resolution structures of both TORC1 and TORC2. However, there is some evidence that suggests that a region in the N-terminal half of Tor2, which based on the low resolution cryo-EM map interacts with the other Tor2 in the dimer, as well as with Avo2 and Avo3, is important for the stable interaction of Tor2 in TORC2 [80]. Inserting this region, denominated major assembly specificity (MAS) domain, into Tor1 creates a chimera which is capable of associating with TORC2, but not TORC1. Future high resolution structures should help in better understanding the structural differences between Tor1 and Tor2 that impact in their function and differential complex assembly.”
Besides these few amendments, I would like to suggest a change in wording. Rapamycin is often called “antibiotic” (line 27). This is certainly correct in the broad sense of the term “antibiotic”. However, in a more narrow sense, “antibiotic” is commonly used as a synonym for “antibacterial”. Therefore, the use of “antibiotic” in the context of rapamycin may be misleading.
To avoid any confusion, we have eliminated “antibiotic” from the sentence and it now reads: “The TOR (Target Of Rapamycin) genes were first identified in a screen of yeast mutants that were resistant to the macrolide rapamycin…”
On a final note, what could be added to this very thorough review are some thoughts on the evolutionary conservation (but this is probably beyond the scope).
We agree with the reviewer that the evolutionary conservation of the TOR pathway is interesting, but it is beyond the scope of the review, which is focused on structural aspects of TOR and PIKKs.
Reviewer 2 Report
In their review “Structural Insights into TOR Signaling” Tafur, Kefauver and Loewith discuss the current structural knowledge on (m)TORC1 and its regulators, as well as (m)TORC2. They present a comprehensive overview of the available data and discuss molecular mechanisms implied by structural analyses of components of the mTOR network.
The review covers an interesting topic and is well written. The reviewer has only minor comments and suggestions to further improve the review:
Comments and suggestions:
- Many abbreviations are not introduced. Please provide a list of abbreviations and/ or an introduce abbreviations when first used in the text.
- The authors introduce (m)TORC2 as rapamycin-insensitive (Page 1 – line 37). While short-term rapamycin treatment only inhibits mTORC1, long term treatment with rapamycin can also affect mTORC2 (e.g. PMID: 16603397). The reviewer suggests that this mode of inhibition is introduced, for instance where the structural basis for rapamycin insensitivity of mTORC2 is discussed (e.g. page 15 – line 457).
- On page 2 – lines 48 to 51, the authors introduce processes regulated by mTORC1 (protein synthesis, nucleotide synthesis, lipid synthesis, lysosome biogenesis, …). The reviewer suggests to introduce relevant citations after each process.
- Figure 1:
- 4E-BP1 inhibits 5´cap dependent translation. However, an activating arrow is depicted in the Figure. Please correct.
- RagA(B) are depicted in blue. The Figure legend states that blue is the colour for negative regulators of mTORC1 and mTORC2. As RagA(B) mediate mTORC1 lysosomal localization, another colour should be used.
- DEPTOR is not shown to inhibit mTORC2. However, DEPTOR inhibits both mTORC1 and mTORC2 (PMID: 19446321).
- The text references Figure 2D before 2C. The reviewer suggests to change the order of the Figure panels such that they appear in order.
- Table 1: mLST8 and DEPTOR appear twice in the table as they are in complex 1 and 2. The reviewer suggests to add an additional section “components of mTORC1 and 2” to avoid redundancy.
- Page 14 – Line 419: Citation 78 does not exist. Please check the references throughout for formatting mistakes.
- Please use HUGO gene nomenclature to support consistent nomenclature in our field.
- Consider citing and discussing the following preprint: https://www.researchsquare.com/article/rs-36453/v1
Alexander Heberle and Kathrin Thedieck, Lab for Metabolic Signaling, University of Innsbruck, Austria
Author Response
Reviewer 2
In their review “Structural Insights into TOR Signaling” Tafur, Kefauver and Loewith discuss the current structural knowledge on (m)TORC1 and its regulators, as well as (m)TORC2. They present a comprehensive overview of the available data and discuss molecular mechanisms implied by structural analyses of components of the mTOR network.
The review covers an interesting topic and is well written. The reviewer has only minor comments and suggestions to further improve the review:
We thank the reviewer for the positive comments.
Comments and suggestions:
- Many abbreviations are not introduced. Please provide a list of abbreviations and/ or an introduce abbreviations when first used in the text.
We have now checked thoroughly in the text and all abbreviations are introduced the first time there are used. However, for some of the proteins mentioned (e.g. subunits of GATOR1, PIKKs), we prefer to not write the full names of the proteins to avoid make reading too difficult, specially considering since the discussed information is already dense.
- The authors introduce (m)TORC2 as rapamycin-insensitive (Page 1 – line 37). While short-term rapamycin treatment only inhibits mTORC1, long term treatment with rapamycin can also affect mTORC2 (e.g. PMID: 16603397). The reviewer suggests that this mode of inhibition is introduced, for instance where the structural basis for rapamycin insensitivity of mTORC2 is discussed (e.g. page 15 – line 457).
We have now added the following sentence and reference (Line 487):
“It is worth mentioning that long-term rapamycin treatment can inhibit mTORC2 signaling in some cell types [101], probably due to the binding of free mTOR molecules prior to their complex assembly.”
- On page 2 – lines 48 to 51, the authors introduce processes regulated by mTORC1 (protein synthesis, nucleotide synthesis, lipid synthesis, lysosome biogenesis, …). The reviewer suggests to introduce relevant citations after each process.
We have now referenced each statement.
- Figure 1:
- 4E-BP1 inhibits 5´cap dependent translation. However, an activating arrow is depicted in the Figure. Please correct.
- RagA(B) are depicted in blue. The Figure legend states that blue is the colour for negative regulators of mTORC1 and mTORC2. As RagA(B) mediate mTORC1 lysosomal localization, another colour should be used.
- DEPTOR is not shown to inhibit mTORC2. However, DEPTOR inhibits both mTORC1 and mTORC2 (PMID: 19446321).
We have now corrected Figure 1 with the indicated suggestions. The color of RagA (blue navy) is consistent with the other figures, and is clearly distinguishable from the light blue used for the boxes of negative regulators. We have now updated the figure legend to explicitly mention each color.
- The text references Figure 2D before 2C. The reviewer suggests to change the order of the Figure panels such that they appear in order.
We have now changed panels 2C and D to fit it to the order in the text.
- Table 1: mLST8 and DEPTOR appear twice in the table as they are in complex 1 and 2. The reviewer suggests to add an additional section “components of mTORC1 and 2” to avoid redundancy.
We have now split Table 1 into two, separating TOR complexes from the TORC1 regulators. We prefer to keep redundant subunits under each complex as it makes it easier for comparison.
- Page 14 – Line 419: Citation 78 does not exist. Please check the references throughout for formatting mistakes.
We apologize for this error and we have now corrected the reference.
- Please use HUGO gene nomenclature to support consistent nomenclature in our field.
HUGO gene names, where different than the protein names, have been added to tables 1 and 2.
- Consider citing and discussing the following preprint: https://www.researchsquare.com/article/rs-36453/v1
We thank the reviewer for the suggested reference. We have now updated the text to include it (Line 336):
“A study currently under review has revealed, by cryo-EM, that the TSC complex consists of two copies of TSC1 and TSC2 each, and only one copy of TBC1D7 [55]. The functional insights derived from this structure are consistent with the previous TSC2 GAP structure [54].”
Reviewer 3 Report
Recent advances in cryo-electron microscopy have facilitated the high-resolution structure determination of macromolecular complex, including TOR complex 1 (TORC1) and TORC2, as well as the TORC1 regulators. This manuscript nicely summarizes the structural studies of the TOR pathway components that have led to the elucidation of the regulatory mechanisms of TORC1 and TORC2 signaling. Moreover, the authors illustrate the similarity in the regulatory mechanisms between TOR and the other phosphatidylinositol 3-kinase-related kinases (PIKKs), such as DNA-PKcs, ATM, ATR, SMG1 and TRRAP. Each section is well-organized, and the manuscript, including figures, is of high quality. This review article would be very helpful for researchers both inside and outside of the TOR field to catch up with the recent rapid progress in the structural biology of TOR signaling.
Minor point:
Table 1 is a little confusing. It might be helpful to prepare a separate table for each complex, rather than one continuous table. In addition, the molecular weight of mTORC1 does not seem to include those of the PRAS40 and DEPTOR. Lastly, some protein names appear in bold and an explanation for it is necessary.
Author Response
Reviewer 3
Recent advances in cryo-electron microscopy have facilitated the high-resolution structure determination of macromolecular complex, including TOR complex 1 (TORC1) and TORC2, as well as the TORC1 regulators. This manuscript nicely summarizes the structural studies of the TOR pathway components that have led to the elucidation of the regulatory mechanisms of TORC1 and TORC2 signaling. Moreover, the authors illustrate the similarity in the regulatory mechanisms between TOR and the other phosphatidylinositol 3-kinase-related kinases (PIKKs), such as DNA-PKcs, ATM, ATR, SMG1 and TRRAP. Each section is well-organized, and the manuscript, including figures, is of high quality. This review article would be very helpful for researchers both inside and outside of the TOR field to catch up with the recent rapid progress in the structural biology of TOR signaling.
We thank the reviewer for his comments.
Minor point:
Table 1 is a little confusing. It might be helpful to prepare a separate table for each complex, rather than one continuous table. In addition, the molecular weight of mTORC1 does not seem to include those of the PRAS40 and DEPTOR. Lastly, some protein names appear in bold and an explanation for it is necessary.
We have taken the reviewer’s suggestion and split the table into two: one showing the TOR complexes and the other one the TORC1 regulators. We have also added additional information in the table legend to improve clarity.